# Study protocol for THINK: a multinational open-label phase I study to assess the safety and clinical activity of multiple administrations of NKR-2 in patients with different metastatic tumour types

Caroline Lonez,[1] Bikash Verma,[2] Alain Hendlisz,[3] Philippe Aftimos,[3] Ahmad Awada,[3] Eric Van Den Neste,[4] Gaetan Catala,[4] Jean-Pascal H Machiels,[4] Fanny Piette,[5] Jason B Brayer,[6] David A Sallman,[6] Tessa Kerre,[7] Kunle Odunsi,[8] Marco L Davila,[6,9] David E Gilham,[1] Frédéric F Lehmann[1]

For numbered affiliations see end of article.

**Correspondence to**
Dr Caroline Lonez;
clonez@celyad.com

## ABSTRACT

**Introduction** NKR-2 are autologous T cells genetically modified to express a chimeric antigen receptor (CAR) comprising a fusion of the natural killer group 2D (NKG2D) receptor with the CD3ζ signalling domain, which associates with the adaptor molecule DNAX-activating protein of 10 kDa (DAP10) to provide co-stimulatory signal upon ligand binding. NKG2D binds eight different ligands expressed on the cell surface of many tumour cells and which are normally absent on non-neoplastic cells. In preclinical studies, NKR-2 demonstrated long-term antitumour activity towards a breadth of tumour indications, with maximum efficacy observed after multiple NKR-2 administrations. Importantly, NKR-2 targeted tumour cells and tumour neovasculature and the local tumour immunosuppressive microenvironment and this mechanism of action of NKR-2 was established in the absence of preconditioning.

**Methods and analysis** This open-label phase I study will assess the safety and clinical activity of NKR-2 treatment administered three times, with a 2-week interval between each administration in different tumour types. The study will contain two consecutive segments: a dose escalation phase followed by an expansion phase. The dose escalation study involves two arms, one in solid tumours (five specific indications) and one in haematological tumours (two specific indications) and will include three dose levels in each arm: $3 \times 10^8$, $1 \times 10^9$ and $3 \times 10^9$ NKR-2 per injection. On the identification of the recommended dose in the first segment, based on dose-limiting toxicity occurrences, the study will expand to seven different cohorts examining the seven different tumour types separately. Clinical responses will be determined according to standard Response Evaluation Criteria In Solid Tumors (RECIST) criteria for solid tumours or international working group response criteria in haematological tumours.

**Ethics approval and dissemination** Ethical approval has been obtained at all sites. Written informed consent will

## Strengths and limitations of this study

► Standard dose escalation phase and an expansion phase to investigate for the first time the safety and initial clinical activity of NKR-2 (natural killer group 2D (NKG2D)-based chimeric antigen receptor (CAR) T cells retrovirally modified) in a multiple injection schedule.

► Phase I study exploiting the multiple ligand targeting capability of human NKG2D by assessing the safety and initial clinical activity of NKR-2 against a broad range of tumour indications, including haematological and solid tumour types.

► Based on the unique NKR-2 mode of action, NKR-2 will be administered without any prior preconditioning.

be taken from all participants. The results of this study will be disseminated through presentation at international scientific conferences and reported in peer-reviewed scientific journals.

**Trial registration number** NCT03018405, EudraCT 2016-003312-12; Pre-result.

## INTRODUCTION

### CAR T cell therapy

The chimeric antigen receptor (CAR) is an artificial construct that endows T cells with predetermined specificity.[1] Prior to adoptive transfer, they have to be expanded in vitro to reach therapeutically sufficient numbers. In this manner, large numbers of tumour antigen-specific T cells can be rapidly generated for use in adoptive cell therapy. The CAR itself has developed through several generations but

has largely followed the configurations of extracellular antigen-binding domain (most commonly a single-chain variable fragment (scFv) derived from an antibody), a transmembrane domain and intracellular signalling domains. At present, second-generation formats tend to predominate featuring a co-stimulatory signalling domain (eg, CD28, 4-1BB) fused to a T cell-activating domain (eg, CD3ζ).[1 2]

The CD19-specific CAR provides the paradigm against which other CAR approaches are currently measured due to the impressive clinical responses seen in early phase clinical trials targeting B cell malignancies. Reports of initial objective clinical responses in 70–90% of patients with advanced chemoresistant B cell leukaemia have fuelled interest in the area with the number of CD19-targeted CAR T cell trials rapidly rising within the last few years.[3 4] The CD19-CAR T cell therapy approach is based on a '*transplant procedure-like*' approach, that is, a single CAR T cell infusion in preconditioned (lymphodepleted) patients and for which persistence, expansion and survival of the transferred T cells are crucial.[3 4] All preconditioning regimens follow a similar concept where there is a transient depletion of circulating lymphoid cell populations to provide 'space' for the adoptively transferred T cells to occupy.[5 6] Additional beneficial effects of preconditioning include the production of homeostatic cytokines and dampening of suppressive factors such as regulatory T cells, thereby providing a window of immunological opportunity to target the tumour.

However, single-target therapy may select for and spur escape variants by clonal selection or antigen loss. Several CD19-specific therapies indeed evidenced cases of relapse resulting from CD19 antigen loss in leukaemia cells post-CD19-specific CAR T cell therapy.[7–10] Furthermore, these therapies are also associated with significant levels of toxicity,[11 12] including cytokine release syndrome (CRS), neurotoxicity and on-target off-tumour toxicity against normal cells (eg, CD19-specific CAR T cells also target normal B cells expressing CD19, causing prolonged B cell aplasia and hypogammaglobulinaemia[13 14]). Moreover, anaphylaxis reaction to the foreign components of the CAR (eg, most CARs use scFv from murine origin, which might potentially induce transgene-specific humoural and cellular immune responses against the murine domain[15–17]) will impact drastically the safety of multiple injection schedule.[18] The risk/benefit is clearly biased towards benefit since these patients have highly advanced cancer with little, if any, alternative effective therapeutic option. However, this toxicity does provide a level of patient selection that impacts recruitment onto current trials and the wider applicability for the general patient population.

By contrast, outside of the B cell malignancy setting, the CAR T cell approach has met with much reduced levels of clinical success.[2] Notably, apart from CD19, there are only a few examples of normal cell lineages that can be safely eliminated along with malignant cells that share the targeted antigen, even within haematological malignancies. Targeting an antigen on tumour cells that is also present on normal cells, for example, on the myeloid lineage, could lead to on-target off-tumour toxicity and serious side effects related to therapy.[19] Early studies targeting solid tumours reported little evidence of clinical effect while there was some evidence of on-target, off-tumour toxicity seen using CAR targeting carbonic anhydrase-IX[16 20] in renal cell carcinoma or Her2/neu-specific CAR T cells in colorectal cancer (CRC).[21] Investigators are therefore still seeking for the ideal target antigen that should be highly overexpressed on tumour cells compared with normal cells. While some recent studies have shown limited clinical effect but nowhere near to that seen with CD19 CAR T cell therapy,[3 22] the challenges posed by solid tumours are numerous and have been extensively reviewed recently.[23–26] In brief, these include the on-target off-tumours impact related to expression of the target on normal tissues, the need for trafficking into the tumour microenvironment (TME) and tumour stroma, tumour heterogeneity and the requirement to overcome the hostile immune suppressive TME, potentially leading to CAR T cell exhaustion, clearly requiring the further development of the basic CAR approach.[27] Several options, including third-generation and fourth-generation CARs (eg, Armoured CAR, T cells redirected for universal cytokine-mediated killing (TRUCKs)), logic gated activation strategies or the combination of CAR T cells with T cell checkpoint inhibitors, are currently under development to solve these problems.

## NKR-2

NKR-2 are autologous T cells genetically modified to express a CAR comprising a fusion of the full-length natural killer group 2D (NKG2D) receptor with the CD3ζ signalling domain. NKG2D is a C-type, lectin-like, type II transmembrane glycoprotein-activating receptor expressed in humans on natural killer (NK), natural killer T, activated CD8[+] T cells and, under certain conditions, some CD4[+] and γδ+T cell subsets[28] and functioning to target and kill cancer or infected cells. In humans, NKG2D is expressed as a homodimer that associates with the adaptor molecule DNAX-activating protein of 10 kDa (DAP10) to form a hexameric structure.[29] The interaction with DAP10 is essential to stabilise the surface expression of NKG2D in T cells and to mediate signal transduction and cellular activation upon ligand recognition.[30] Therefore, while the NKR-2 CAR format suggests a 'first-generation' configuration, the NKG2D structural component, via its interaction with endogenous DAP10, engages native T cell co-stimulatory signalling apparatus to providing a sufficient co-stimulatory signal which complements the primary CD3ζ vector-derived signal, indicating that the NKR-2 actually works rather like a 'second-generation' CAR (figure 1). Furthermore, the requirement for endogenous DAP10 expression for stable expression of NKG2D within cells defines an additional embedded safety measure to restrict NKR-2 vector expression and function specifically to cells expressing native DAP10 (eg, T cells).

In humans, eight NKG2D ligands have been identified, encoded by different loci on two different arms of the

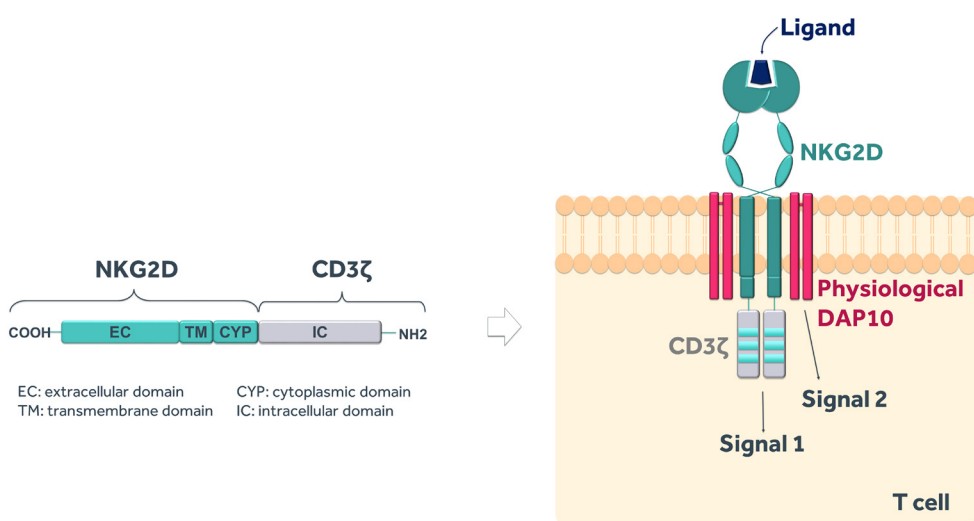

**Figure 1** The NKR-2 construct. NKR-2 design contains the full-length human natural killer group 2D (NKG2D) receptor linked to the signalling domain of CD3ζ that provides primary signalling (signal 1) to activate T cells upon ligand binding. The naturally expressed adaptor molecule DNAX-activating protein of 10 kDa (DAP10) provides secondary signalling (signal 2) which allows NKR-2 to work as a second-generation CAR.

chromosome 6 and corresponding to two main families: the major histocompatibility complex (MHC) class I chain-related proteins (MIC) which include MICA and MICB proteins, and the structurally diverse unique long 16-binding proteins (UL16-binding proteins or ULBP) which include ULBPs 1 to 6.[31] The expression of these ligands can be induced by stress such as viral infection, oxidative or thermal stress, genotoxic drugs, DNA-damaging agents, tissue damage, heat shock, inflamed tissues (eg, autoimmune diseases) and malignant transformation.[32] Importantly, a broad range of primary tumours express NKG2D ligands.[33 34] While the NKG2D ligand expression profile is often heterogeneous, the large majority of tumour samples express at least one ligand (and often multiple ligands), including haematological and solid tumour types, implying that the profile of ligand expression is suitable to be exploited as a cancer therapy.[34] Finally, the surface expression of NKG2D ligands can be modulated by proteolytic cleavage with the release of the soluble ectodomain.[35 36] These shed NKG2D ligands may downmodulate NKG2D expression on NK cells and T cells, contributing to tumour immune escape and elevated serum levels were accordingly often associated with poor prognosis.[35 36]

The breadth of ligands targeted by NKG2D provides T cells armed with the NKR-2 receptor the potential to challenge an extensive range of tumour indications.[33 34]

Preclinical studies have shown the murine equivalent of NKR-2 (chNKG2D T cells) can mediate potent antitumour activity against both haematological and solid tumours,[37] including highly heterogeneous tumours with less than 10% ligand-positive tumour cells (see table 1 for a summary of these data).[38]

Critically, these preclinical data in multiple syngeneic tumour models indicate that antitumour efficacy was not dependent on lymphodepleting preconditioning, while the induction of a tumour-specific adaptive immune response by NKR-2 therapy enabled the long-term protective effects of the approach.[39] Recent data also confirmed that human NKR-2 are effective in vivo in a mouse xenograft model of human pancreatic cancer.[40] Importantly, preclinical data suggest four different mechanisms of action of NKR-2, including (i) direct cytotoxicity against cancer cells, (ii) anti-angiogenic activity potentially through targeting of NKG2D ligands expressed on tumour neovasculature,[41] (iii) modulation of the immune TME through direct targeting of immunosuppressive myeloid-derived suppressor cells and regulatory T cells (Tregs), and recruitment of myeloid cells and activated macrophages,[42 43] together resulting in (iv) the observed generation of a tumour-specific adaptive immune response[42–48] (figure 2). Preclinical data also demonstrated that NKR-2 rapidly disappeared from tissues by 7 days after injection suggesting a short-term persistence of NKR-2 (without uncontrolled expansion), although being able to induce a long-lasting memory host immune response against tumour-specific antigens.[39] Finally, preclinical data demonstrated that NKR-2 remained functionally intact in the presence of soluble MICA at concentrations much higher than the physiological concentrations.[49]

Expressing the NKR-2 CAR into T cells therefore allows the NKR-2 cells to present the broad specificity characteristic of NK cells while exploiting the migration, expansion, long half-life and potential to generate immunological memory characteristic of T cells. In addition to direct recognition of tumour cells, targeting of ligands expressed on the stroma of the TME can also lead to disruption of the essential support mechanisms required for tumour cell survival and growth. Moreover, in contrast to other single or limited target CAR T cell therapies, the fact that NKR-2 targets several ligands from two main

**Table 1** Preclinical studies with NKR-2

| | Targeted model | Publications | Main findings |
|---|---|---|---|
| **In vitro experiments** | | | |
| Murine equivalent of NKR-2 | Lymphoma | Zhang et al, 2007[44], Zhang et al, 2005[60] | NKR-2 induces interferon (IFN)-γ expression, cytokine release upon co-culture with tumour cell lines and triggers specific cytotoxicity against several tumour types |
| | Myeloma | Barber et al, 2011[39] | |
| | Ovarian cancer | Barber et al, 2008b[45], Spear et al, 2013b[46], Barber et al, 2007[61] | |
| | Melanoma | Zhang and Sentman, 2013[41] | |
| Human NKR-2 | Myeloma | Barber et al, 2008a[62] | NKR-2 induces IFN-γ expression, cytokine release upon co-culture with tumour cell lines and triggers specific cytotoxicity against several tumour types |
| | Tumour cell lines | Demoulin et al, 2017[40], Zhang et al, 2006[49] | |
| **In vivo experiments** | | | |
| Murine equivalent of NKR-2 | Lymphoma-bearing C57BL/6 mice | Spear et al, 2013a[38], Zhang et al, 2007[44], Zhang et al, 2005[60] | Efficacy of single intravenous and subcutaneous administration of NKR-2 (1–7.5×10$^6$ cells/injection) and resistance to a rechallenge with same tumour type |
| | | Spear et al, 2013a[38] | Efficacy against heterogeneous tumours |
| | | Zhang et al, 2007[44] | Efficacy of multiple intravenous administration of NKR-2 (5×10$^6$ cells/injection) |
| | | Sentman et al, 2016[54] | Single and multiple intravenous injections of NKR-2 (5×10$^6$–2×10$^7$ cells/injection) are not toxic at doses <2×10$^7$ NKR-2 per injection |
| | Multiple myeloma-bearing C57BL/KaLwRij mice | Barber et al, 2011[39] | Efficacy of single and multiple intravenous administration of NKR-2 (5×10$^6$ cells/injection) and resistance to a rechallenge with same tumour type; Effect of lymphodepletion; Host memory immune response; Low persistence of NKR-2 in the body after administration |
| | Ovarian tumour-bearing C57BL/6 mice | Barber et al, 2009[42], Spear et al, 2012[43], Barber et al, 2008b[45], Spear et al, 2013b[46], Barber and Sentman, 2009[47]; Barber et al, 2007[61] | Efficacy of single intraperitoneal administration of NKR-2 (5×10$^6$ cells/injection) and resistance to a rechallenge with same tumour type |
| | | Barber et al, 2009[42], Spear et al, 2012[43] | Efficacy against established tumours |
| | | Spear et al, 2012[43], Barber et al, 2008b[45], Spear et al, 2013b[46], Barber and Sentman, 2009[47], | Effect on tumour microenvironment; Host memory immune response |
| | | Spear et al, 2013a[38], Barber et al, 2009[42], Barber et al, 2008b[45], Spear et al, 2013b[46] | Efficacy of multiple intraperitoneal administration of NKR-2 (5×10$^6$ cells/injection) |
| | | Spear et al, 2013a[38] | Efficacy against heterogeneous tumours |
| | Melanoma-bearing C57BL/6 mice | Zhang and Sentman, 2013[41] | Efficacy of multiple intra tumoural administration of NKR-2 (2×10$^6$ cells/injection); Effect on tumour vasculature |
| | Healthy C57BL/6 mice | Sentman et al, 2016[54] | Single and multiple intravenous injections of NKR-2 (5×10$^6$–2×10$^7$ cells/injection) are not toxic at doses <2×10$^7$ NKR-2 per injection. |
| Human NKR-2 | Pancreatic tumour xenograft-bearing NOD-scid IL2Rgamma$^{null}$ mice | Demoulin et al, 2017[40] | Efficacy of multiple intraperitoneal administration of NKR-2 (5×10$^6$ cells/injection) |

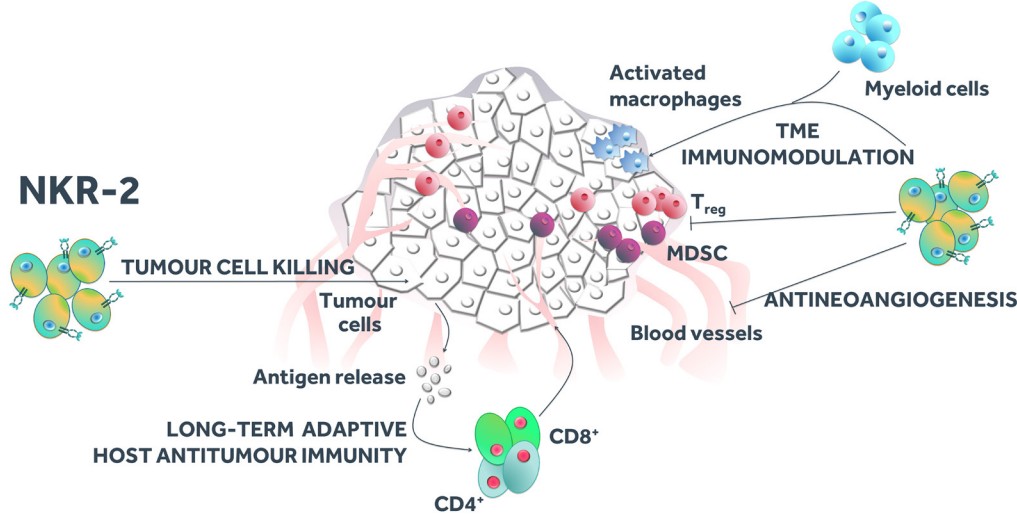

**Figure 2** NKR-2 different mechanisms of action. Apart from direct cytotoxicity against cancer cells, NKR-2 mode of action also involves reduction of blood vessel density indicative of an antiangiogenic activity, modulation of the immunosuppressive tumour microenvironment (TME) and induction of a long-term antitumour-specific memory immune response. MDSC, myeloid-derived suppressor cell.

families putatively decreases the probability of target loss and associated relapse after treatment. Further, the lack of important in vivo cell expansion and long-term survival of NKR-2, and the induction of tumour-specific adaptive immunity in the absence of patient preconditioning, also bears a safety advantage preventing side effects generally associated with rapid CAR T cell expansion[50] which often require development of strategies like embedded suicide systems or use of chemotherapy to eradicate the CAR T cells.[51 52] Collectively, this suggests that exploiting the targeting of NKG2D ligands by NKR-2 to fight against diverse tumour indications with a single generic construct combining two features of innate and adaptive immunity provides potentially a new paradigm for CAR T cell therapy that the current protocol is designed to investigate.

## METHODS AND ANALYSIS
### Study objectives
The THerapeutic Immunotherapy with NKR-2 (THINK) trial is an open-label phase I study which primarily aims to assess the safety and clinical activity of the NKR-2 treatment administered three times every 2 weeks between each administration in different tumour types. The study is split into two segments each with specific study objectives. The first dose escalation segment focuses on determining the recommended dose (RecD) of NKR-2 cells for second segment of THINK. Segment 2 extends the safety study and also investigates initial clinical activity of NKR-2 across multiple tumour indications. In both segments, correlative measures of clinical activity will be examined.

### Study drug
NKR-2 refers to the viable autologous CD3⁺ cells obtained after transduction of autologous T cells with the NKG2D-based CAR, which contains a total viable population of at least 80% CD3⁺ cells, with at least 50% CD3⁺ being transduced. NKR-2 will be supplied cryopreserved in bags containing a T cell dose in accordance with the dose level which is to be administered.

### Rationale for the NKR-2 dose and regimen
NKR-2 has been used in the single infusion dose escalation clinical study CM-CS1 (NCT02203825) which evaluated the safety and clinical activity of NKR-2 administration without prior lymphodepleting conditioning in patients with two distinct haematological malignancies: acute myeloid leukaemia (AML)/higher risk myelodysplastic syndrome (MDS) refractory anaemia with excess blasts, that are not in remission and for which standard therapy options are not available, and relapsed or refractory progressive multiple myeloma (MM) after several lines of standard therapy. The trial has been successfully completed with no observed dose-limiting toxicities (DLTs) at the highest dose infused ($3 \times 10^7$ total cells) and initial signs of clinical benefit.[53]

In preclinical models, the pharmacologically effective dose (PED) of the murine NKR-2 ranged from $1 \times 10^6$ cells to $7.5 \times 10^6$ cells per dose (intravenous single injection). Although a significant antitumour activity was obtained with a single injection of murine NKR-2, 100% overall survival and complete recovery were obtained following repeated three sequential doses of $5 \times 10^6$ cells/mouse without any adverse effect.[39 42 44 45] Moreover, a treatment regimen with 2 weeks' interval between injections was far more successful in terms of activity compared with a weekly interval demonstrating that successful treatment of established tumours did not require the administration of more NKR-2.[42] The maximum tolerated dose (MTD) after a single intravenous injection was determined to be

$1 \times 10^7$ cells, and repeated administration of this dosage to healthy and tumour-bearing mice did not produce any major toxicities.[54]

Following these values of PED and MTD, the no-observed effect level was defined at $5 \times 10^6$ cells/injection per mouse and were identical whatever the tumour-type model (haematological or solid tumour). Using the allometry scaling exponent of 0.75, as defined by Food and Drug Administration (FDA) guidelines (USFDA, 2005), human-equivalent PED and MTD were therefore estimated at $0.43$–$3.2 \times 10^9$ and $4.3 \times 10^9$ cells/injection, respectively. Together these observations of high tolerability in initial single dose phase I clinical trial coupled with preclinical efficacy experimentation strongly support the concept of a dose escalation segment of $3 \times 10^8$–$3 \times 10^9$ NKR-2 per infusion given three times each separated by a 14-day interval.

## Patient population

Refractory or relapsing patients with metastatic or locally advanced CRC, urothelial carcinoma, triple-negative breast cancer (TNBC), pancreatic cancer, recurrent epithelial ovarian and fallopian tube carcinoma, AML/MDS and MM, which failed from standard treatments, will be recruited into this protocol (the key inclusion and exclusion criteria are presented in box).

The study will enrol patients in different countries in Europe and in USA with a dose escalation segment open to a restricted academic network in Belgium and USA. List of study sites can be obtained on clinicaltrials.gov (NCT03018405).

The tumour types targeted in this protocol were selected primarily based on their high expression of the NKG2D ligands (with a prevalence of 78–100% tumour samples expressing at least one ligand).[33 34] Furthermore, NKG2D ligand expression is a dynamic and context-dependent process, quickly varying over time in response to cytokines in the TME and evolving along with tumour progression, local inflammatory responses and treatments,[33 34] hence difficult to assess. Finally, even if NKG2D ligand expression is heterogeneous with respect to the exact ligands found, based on preclinical data, potential therapeutic benefit might be achieved in patients with highly heterogeneous tumours[38] or even tumours considered as 'negative' for NKG2D ligands because of a NKG2D ligand-positive TME and/or neovasculature and/or activation of host immune system.[42–48] Thus, no eligibility criteria were based on the individual expression profile of NKG2D ligands and no diagnostic test for ligand expression was used for recruitment into this study. Correlative studies, however, will be carried out during the trial and will attempt to investigate the expression profile of patient tumours with clinical response.

## THINK study design

THINK phase I study will contain two consecutive segments: a dose escalation and an expansion segment (figure 3).

---

**Box    Key eligibility criteria**

**Main inclusion criteria:**
► Men or women ≥18 years old at the time of signing the informed consent form (ICF).
► Patient with specific cancer indications (see below).
► Disease must be measurable according to the corresponding guidelines.
► Eastern Cooperative Oncology Group (ECOG) performance status of 0 or 1 or 2 based on peripheral neuropathy from prior therapies.
► Patient with adequate bone marrow reserve, hepatic and renal functions.
► Left ventricular ejection fraction of >40%.
► For patients with solid tumour type, the patient must agree to have a tumour biopsy at baseline.

**Main exclusion criteria:**
► Patient with a tumour metastasis in the central nervous system.
► Patients who have received another cancer therapy within 2 weeks before the planned day for the apheresis.
► Patients who receive or are planned to receive any other investigational product within the 3 weeks before the planned day for the first NKR-2 administration.
► Patients who are planned to receive concurrent growth factor, systemic steroid or other immunosuppressive therapy or cytotoxic agent.
► Patients who underwent major surgery within 4 weeks before the planned day for the first NKR-2 administration.
► Patients who have active infections necessitating the use of antibiotics/antivirals treatment.
► Patients with history of autoimmune disease.

**Disease-specific inclusion criteria**
**Patients must have either:**
► A documented metastatic or locally advanced colorectal adenocarcinoma and having received, being intolerant or having refused at least two prior standard cancer therapy regimens as part of their primary treatment regimen or part of their treatment for management of recurrent/persistent disease.
► A documented recurrent epithelial ovarian cancer or fallopian tube carcinoma and having received or refused at least two prior standard cancer chemotherapy regimens as part of their primary treatment regimen or part of their treatment for management of recurrent/persistent disease.
► An inoperable locally advanced or metastatic urothelial carcinoma that has progressed after previous platinum-based chemotherapy.
► A metastatic or locally advanced triple-negative breast cancer and having received at least one prior cancer therapy regimen as part of their treatment for management of recurrent/persistent disease.
► A metastatic or locally advanced pancreatic ductal adenocarcinoma and having received, being intolerant or having refused at maximum two prior lines of standard cancer chemotherapy regimens.
► Relapsed or refractory acute myeloid leukaemia (ie, ≥5% blasts in bone marrow or in peripheral blood) after one prior therapy or patients with myelodysplastic syndrome (ie, ≥5% blasts in bone marrow or ≥2% blasts in peripheral blood) or MDS with TP53 mutation who failed from prior treatments.
► Relapsing or refractory/relapsing multiple myeloma with a minimum of one prior line of systemic therapy.

---

The *dose escalation* design will include three dose levels of NKR-2. The assumption is made that the safety profile of the treatment could be different in haematological

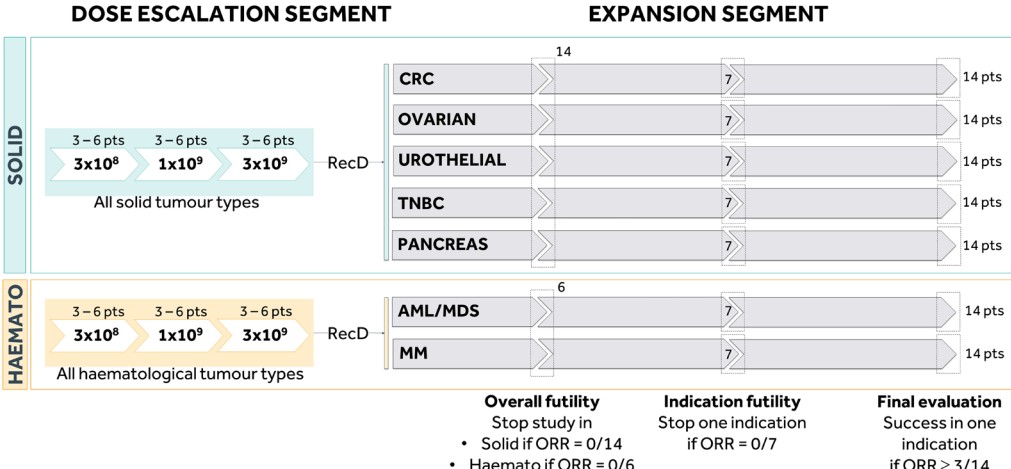

**Figure 3** Overview of the study design. AML, acute myeloid leukaemia; CRC, colorectal cancer; MDS, higher risk myelodysplastic syndrome; MM, multiple myeloma; ORR, objective response rate; RecD, recommended dose; TNBC, triple-negative breast cancer.

versus solid tumour types. Therefore, it will be assessed separately in two different arms, one in solid tumours (CRC, urothelial carcinoma, TNBC, pancreatic cancer or epithelial ovarian and fallopian tube carcinoma) and one in haematological tumours (AML/MDS or MM). Each arm will use a 3+3 design to determine the RecD of the NKR-2 treatment for the corresponding cohorts in the expansion segment, as based on the occurrence of DLTs (ie, three patients will be evaluated per dose level for each arm of the segment and three additional patients will be added if one out of three patients in this cohort experiences a DLT to further assess the safety of the treatment). The sample size of the dose escalation part will be between 2 and 18 patients per arm. Six patients will be treated at the MTD (or highest dose level if no DLT was observed).

The RecD will be the MTD unless in case no MTD is determined in the dose escalation segment of the study. In the latter, the RecD will be highest dose evaluated in the dose escalation segment.

The RecD will be further evaluated in the *expansion segment* of the study to assess separately the safety profile and initial clinical activity of the NKR-2 treatment in seven cohorts of patients with the distinct tumour types (up to 14 patients per tumour type). The expansion segment in haematological and solid tumours will be initiated as soon as their respective dose escalation arm defines the RecD. During this segment, the RecD can still be adapted according to predefined safety rules in any specific tumour type. The statistical analysis is planned in three steps. A futility analysis will be conducted separately in the first 14 patients with a solid tumour and the first 6 patients with a haematological tumour type. These number of patients have been defined to have a 0.05 probability of finding 0 objective response at this step if the true probability of response were 0.2 (solid) or 0.4 (haematological). Subsequently, futility (no response in the first seven patients) and efficacy (three or more

responses in 14 patients) will be tested by cohort, based on the Simon's two-stage optimal design. The assumptions are: type I error rate=0.15, power=0.80 and response probability of poor drug=0.10/beneficial drug=0.30. In total, up to 86 patients are expected to be enrolled in the expansion segment (in addition, the six patients from each arm of the dose escalation segment who were treated at the RecD will be included in these expansion analyses).

### Treatment regimen

Three NKR-2 dose levels will be evaluated during this study, that is, $3\times10^8$, $1\times10^9$ and $3\times10^9$ NKR-2 for each injection (adjusted to $4.6\times10^6$, $1.5\times10^7$ and $4.6\times10^7$ NKR-2/kg, respectively, for patients with body weight below or equal to 65 kg) with a schedule of administration of three NKR-2 doses administered with a 2-week interval per patient. No systemic lymphodepleting preconditioning will be performed prior to NKR-2 injections.

### Trial endpoints

The *primary endpoint* of the *dose escalation* segment will be the occurrence of DLTs in all patients during the study treatment until 14 days after the last study treatment administration. *Secondary endpoints* will include the (i) occurrence of adverse events (AEs) and serious adverse events (SAEs) during the study treatment, (ii) occurrence of objective clinical response at D57, M3, M6, M9, M18 and M24 after first NKR-2 administration and according to international guidelines and (iii) progression-free survival (PFS) and overall survival (OS), and duration of response for patients with objective response up to 2 years after first NKR-2 administration.

The *primary endpoint* of the *expansion segment* will be the objective response rate (ORR) at D57, M3, M6, M9, M18 and M24 after first NKR-2 administration. *Secondary endpoints* will include the (i) PFS and OS, (ii) duration of response for patients with objective response up to 2 years

after first NKR-2 administration and (iii) occurrence of AEs and SAEs and DLTs during the study treatment.

The overall aim of this segment is to understand the safety profile and develop initial observations of clinical activity in a cohort of up to 14 patients that support a deeper assessment of clinical efficacy of NKR2 in larger patient cohorts within a phase II clinical study.

Response to treatment will be assessed according to standard Response Evaluation Criteria In Solid Tumors (RECIST) criteria (version 1.1) for solid tumours or international recommendations for response criteria in haematological tumours. Apart from the visits scheduled per protocol, tumour assessment might also be performed at any time if considered as needed per investigator's decision (eg, progression suspicion) and with regard to the RECIST 1.1. response confirmation.

### Study duration

For each patient, the overall study duration will be 15 years. The study consists of a screening phase, a treatment administration phase, a treatment follow-up phase and a long-term safety follow-up (figure 4). The duration of the administration phase will be 8 weeks, from Visit D1 to Visit D57. The duration of the treatment follow-up phase will be 22 months, from the end of the administration phase on D57 (Week 8) to Visit M24.

As per regulatory requirements, patients who received at least one NKR-2 treatment will need to be followed annually for a period of 15 years. Therefore, patients will also follow a *long-term safety follow-up* that will last for up to 15 years after the day of the first NKR-2 administration to collect safety data for 13 additional years. The surveillance will mimic as much as possible the standard

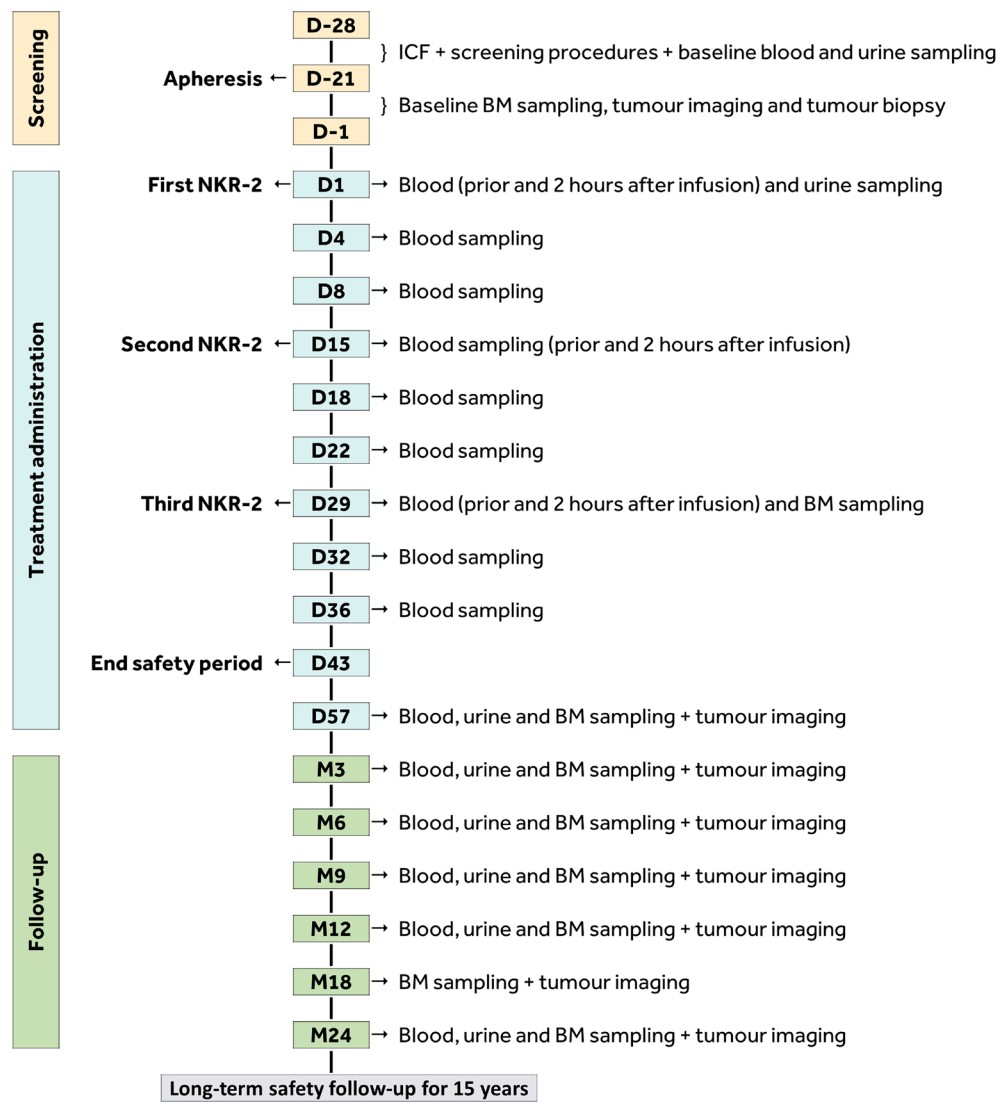

**Figure 4** Overview of study phases. NKR-2 will be infused on days 1, 15 and 29. Tumour assessment will be done by tumour imaging and blood sampling for solid tumours and tumour imaging, blood, urine and bone marrow sampling for haematological tumours. BM, bone marrow; D, day; ICF, Informed Consent Form; M, month.

follow-up of the respective tumour type, including monitoring of important safety parameters.

Patients may be withdrawn from the study treatment or from the study at any time at the patients' discretion or at their investigator's discretion. The investigator will document on the patient's record whether the decision to discontinue treatment and/or study participation was made by the patient or the investigator and which reason applies.

In case of absence of disease progression at M24, PFS and OS as clinical endpoints will be followed up on a yearly basis or at ad hoc *basis*, that is, in case of progression/relapse suspicion. In case of disease progression at M24 or at any time before, the patient will be still asked to attend simplified follow-up procedures, which will include (i) record of survival status, (ii) medication status and (iii) blood sampling for viral copy number (VCN) and replication competent retrovirus (RCR) assessment. Clinical progression at the end of the treatment phase (D57) will not to be considered as a criterion for withdrawal from the study and will allow the continuation of the active follow-up of the patient with or without the initiation of a new anticancer therapy.

### Enrolment rules

There is no requirement for specific recruitment of each individual tumour indication to corresponding arm within the *dose escalation pha*se, but each dose level should enrol at least two different types of cancer. In each arm, patients will be enrolled at the next dose level when all evaluable patients at the same dose level have completed the safety period of 14 days after last NKR-2 injection. Within each dose level, patients will be enrolled as soon as the previous patient has received at least one NKR-2 administration and has completed the 14-day safety period after the first injection.

Enrolment in the *expansion segment* of the study will be performed in three steps, with a statistical analysis at the end of each step (see Adverse events, serious adverse events and dose-limiting toxicities section). Within each step, all patients can theoretically be enrolled at the same time.

Approximately 15 centres are expected to be initiated for patient enrolment in the expansion segment. Recruitment is anticipated to be completed within approximately 24 months. If this target is not met, then the recruitment period may be extended accordingly.

### Adverse events, serious adverse events and dose-limiting toxicities

All AEs starting from registration to 28 days after the last study treatment (Visit D57) must be recorded into the Adverse Event form in the patient's electronic case report form (eCRF), irrespective of intensity or causality with study treatment.

To ensure patient safety, every SAE, regardless of suspected causality, occurring after the patient enrolment and until the administration phase concluding visit (at D57) must be reported to the sponsor within 24 hours of learning of its occurrence. Any SAEs experienced outside this period will only be reported to the sponsor if the investigator suspects a causal relationship to the study participation or if fatal SAEs occur after signature of the Informed Consent Form (ICF).

DLT refers to a toxicity that is experienced during treatment and within 14 days following any NKR-2 dose, is new and at least possibly related to study treatment. This includes any Grade 3 or higher toxicity or Grade 2 or higher autoimmune toxicity that cannot be controlled to Grade 1 or less within 72 hours with appropriate treatment, with the exception of Fever Grade 3 and immediate hypersensitivity reactions occurring within 2 hours of cell infusion that are reversible to a Grade 2 or less within 24 hours of cell administration with standard therapy.

Whenever a patient experiences toxicity that fulfils the criteria for a DLT, the study treatment will be interrupted. The standard time period for collecting and recording DLTs will be 14 days after each study treatment administration. AEs/SAEs that meet the definition of DLTs but that are reported outside of this time window will be reported as AEs/SAEs but not as DLTs.

The standard time period for collecting and recording pregnancy will begin at the first receipt of the product and last until the end of the study.

In addition to the above-mentioned reporting requirements and in order to fulfil international reporting obligations, SAEs that are related to study participation (eg, protocol-mandated procedures, invasive tests, a change from existing therapy) will be collected and recorded from the time the patient consents to participate in the study until she/he is discharged.

### Conduct of analysis and statistical considerations

All statistical analyses will be performed using SAS (SAS Institute) and/or StatXact (Cytel Software Corp) softwares. The results will be reported per dose level/tumour-type cohort and overall.

### Demographic characteristics

Physical examination (age, gender and race), medical history and baseline tumour characteristics will be tabulated and analysed with appropriate descriptive statistics on the *total treated population*.

The *analysis of safety* will be performed on the *total treated population*. All the AEs will be graded according to the National Cancer Institute Common Terminology Criteria for Adverse Events (NCI-CTCAE), Version 4.03 and will be coded to the preferred term level using the Medical Dictionary for Regulatory Activities (MedDRA). A summary of AEs by maximum grade and cohort will be generated. The same analysis will be presented for SAEs and events with a suspected relationship to the study treatment. A summary of AEs by maximum grade and cohort per treatment dose will also be performed. SAEs, DLTs and discontinuation due to AEss will be described in detail by patient narratives.

A summary of the concomitant medications, classified by Anatomical Therapeutic Chemical (ATC) term, will be displayed by cohort. Individual listing of concomitant medications per patient and per time point will also be provided. Finally, the total number of study treatments administered to each patient and the length of active follow-up will be summarised.

The *analysis of clinical activity* will be performed on the *total treated population,* that is, all patients who received at least one NKR-2 treatment (in case of patient's withdrawal prior the end of the administration phase). Sensitivity analyses related to the clinical activity endpoints will be performed on the *according to protocol* population which includes all evaluable patients who have complied with all the procedures defined in the protocol up to concluding visit.

The first ORR will be assessed at Visit D57. The occurrence and duration of objective clinical response Complete response (CR), partial response (PR) and stable disease (SD) will be reported. Disease progression, including death, during the study will be described. Estimates of the rates of best clinical response and their 95% CIs will be reported. Kaplan-Meier curves will be estimated for PFS and OS.

In addition, other exploratory analyses may be performed, *tha*t is, the clinical activity may be assessed in subpopulations based on the NKG2D ligands tumour expression, types of ligands expressed in tumour, serum levels of soluble NKG2D ligands and so on.

In the dose escalation segment, safety data will be monitored closely on an ongoing basis. The main analysis (leading to the selection of the RecD) will be conducted separately in each arm once all patients in the last dose level have received the three NKR-2 administrations and have completed the safety period (D43). All data related to the primary endpoints of the study will be analysed. The final analysis will be performed once all patients have attended the concluding visit of the follow-up phase (M24) or have been withdrawn from the study.

The expansion segment of the study will proceed in three steps, with a statistical analysis at the end of each step (figure 3). These steps aim to stop the study for futility in specific cohorts or specific tumour types in case of no (sufficient) responders in the corresponding cohorts at the time of analysis.

An interim analysis for futility will be performed on the first 14 patients over all 5 cohorts of patients with solid tumour types and on the first 6 patients over the 2 cohorts of patients with haematological tumour types who have either completed the administration phase (D57) or have withdrawn from the study.

The second step consists in cohort futility analyses after the first seven cancer patients in each tumour type have either completed the administration phase (D57) or have withdrawn from the study. At this step, an evaluation of the safety of the NKR-2 treatment in each cohort will also be performed on the first six patients who completed the 6-week treatment period (D43) or had a DLT. This will

allow to adapt the RecD in any specific cohort (tumour type) according to both safety rules and futility analysis. Indeed, if, and as soon as, two or more DLTs have been reported within a specific cohort, and the criterion for futility was not reached, the next patients enrolled in this cohort will be treated at the next lower dose level. In contrast, if no DLT has been reported within a specific cohort, and the criterion for futility was not reached at Step 2, the next patients enrolled will be treated with the next higher dose level.

Finally, the final evaluation of the objective clinical response rate will be evaluated once all the first 14 cancer patients in each cohort have either completed their administration period (D57) or have withdrawn from study due to disease progression or DLT. If, in any cohort, there are >2 out of 14 patients presenting an ORR, the study will be considered as successful in this tumour type.

### Correlative studies

Peripheral blood samples, bone marrow aspirates from patients with haematological malignancies, as well as tumour biopsies from patients with solid malignancies, will be collected at regular time points during the study (see figure 4) and used to determine NKR-2 persistence post-infusion (using a VCN test) and assess the presence of RCR. Systemic cytokine levels in peripheral blood and levels of tumour-related biomarkers (in the serum, bone marrow or urine samples) will also be assessed to determine whether correlates of NKR-2 activity can be identified. Evaluation and characterisation of the NKR-2 mode of action involving the characterisation of the pre-existing and post/during therapy humoural and cellular immune responses will be done on blood samples collected at baseline and post-NKR-2 infusion. Finally, the correlation with the tumour expression of NKG2D ligand in tumour tissues or cells (type and level) and NKR-2 activity, the characterisation of the TME, the NKR-2 infiltration and the modulation of the local tumour environment of this new type of T cell therapy will be evaluated.

### Toxicity management

As previously mentioned, the use of CAR T cells is limited by potentially severe toxicities, including CRS, neurotoxicity and tumour lysis syndrome (TLS), on-target off-tumour toxicity (related to recognition of the target antigen on normal tissues), clonality and insertional oncogenesis, vector persistence and immunogenicity. While these events may have been seen in relation with other CAR T cell therapies, and are provided for information, there are currently no events considered as expected for NKR-2 therapy (which can be different from other CAR T cell studies based on the unique mode of action) and all serious adverse reactions (SARs) will be considered as suspected unexpected SARs (SUSARs).

The management of toxicities related to CAR T cell therapies was described elsewhere[11 12 55] but is evolving with everyday new information available from ongoing studies. Sites will be kept informed all along the study for

any new therapeutic options modifications or standardisation between the different sites.

Briefly, CRS toxicity is currently of highest concern in the field. It is caused by a release of inflammatory cytokines such as interleukin (IL)-6, IL-2, Interferon (IFN)-γ and Tumour Necrosis Factor (TNF)-α, following the recognition of the target antigen, resulting in a systemic inflammatory response similar to sepsis. Therefore, CRS will correlate with both efficacy and toxicity in patients receiving T cell-engaging therapies. Tocilizumab (anti-IL-6) will be used as the first-line agent in the treatment of CRS. If the patient's condition does not improve or stabilise within 24 hours of the tocilizumab dose, administration of a second dose of tocilizumab and/or a second immunosuppressive agent (such as corticosteroids) will be considered, based on the CRS revised grading system created by Lee et al[56] to identifying severity and implementation of optimal care strategies for CRS.

TLS resulting in renal insufficiency, rapidly rising uric acid level or evidence of organ dysfunction will be managed with fluids and rasburicase as needed and as determined by the treating physicians.

Further, as NKG2D ligands may be upregulated on stress signals, expression of ligands on non-cancer cells can be expected in tissues that are actively infected or inflamed or experienced cell stress of other aetiology. Infections could therefore invoke a significant inflammatory response by NKR-2 by potentially inducing a CRS due to the increased recognition of ligands by NKR-2 cells. Therefore, any inflammation, infection, concurrent or previous treatment with anticancer therapies will be carefully registered and any Grade 2 or higher autoimmune toxicity will be closely followed to detect any evidence of on-target off-tumour toxicity. Evidence of acute infection, as well as fever of 38°C/100°F or above, will be therefore considered as an exclusion criterion at screening (see box) or postponement criterion when observed during the treatment phase. In the latter, all study procedures and treatment administration will be resumed where it was left as soon as the patient's condition allows (within 4 weeks after the scheduled treatment date) or at the investigator's discretion after steering committee and sponsor approvals.

Since NKR-2 construct consists entirely of human sequences, it is not expected to result in immunogenicity sometimes observed with other CARs using scFv from murine origin (see section Introduction), and therefore multiple NKR-2 injections are allowed without the anticipated risk of humoural immune responses.

Uncontrolled expansion and proliferation of the transformed cells increase the risks for onset of CRS and other CAR-associated toxicities. However, in contrast to more classical CAR T cell therapies, NKR-2 demonstrated neither significant in vivo expansion nor long-term persistence until now,[39] but since this is the first trial with NKR-2 administered as multiple doses, the persistence of NKR-2 will be nevertheless carefully monitored post-infusion as mandatory exploratory research. If uncontrolled T cell proliferation occurs (Grade 3 or 4 toxicity related

to NRK-2), and as it was reported that CAR-associated toxicity responds to systemic corticosteroids, patients may be treated with corticosteroids and/or chemotherapy to eradicate the CAR T cells.[57] Patients will be treated with pulse, followed by a rapid taper.

Finally, following FDA recommendation for retroviral and lentiviral vector based gene therapy products, the patient monitoring schedule will also include analysis of the RCR of patient samples up to 15 years after CAR T cell injections.

## DISCUSSION

The THINK trial aims primarily to determine the safety and clinical activity of NKR-2 cell therapy, without preconditioning chemotherapy, in patients with advanced cancer. The dose escalation phase of the trial is designed to determine the RecD in haematological and solid tumours, which then is used in the expansion phase to assess clinical responses separately in seven different indications. Due to the multitargeting specificity of NKR-2, the dose escalation and subsequent expansion segments are designed to facilitate a rapid assessment of NKR-2 activity across a range of tumours. This is advantageous since the assessment of safety and any evidence of clinical activity will be suitably evaluated across cancer types using a consistently produced cell product, thereby avoiding issues that arise in multiple single tumour indication trials testing a single T cell product. Interestingly, the number of ligands targeted by NKR-2 (eight ligands from two main families), combined with their stress-induced tissue expression, also limits the potential risk of clonal selection or antigen loss after injection, which has been observed for classical CAR constructs targeting a single antigen. This is expected to translate into limiting the risk for potential relapse.

For patients, the lack of preconditioning chemotherapy removes the need for an inpatient stay during the transient period of lymphodepletion translating to a likely improved patient experience of the trial. Additionally, the absence of preconditioning will also reduce the potential toxicity of this treatment and facilitate providing a therapeutic solution to a larger patient population, including, notably, the elderly population.

The challenge facing classical CAR T cell therapies in the human patient with advanced cancer (beyond B cell malignancies) is major and, in the solid tumour situation, requires the cells to traffic to sites of tumour, overcome the highly suppressive TME and initiate direct tumour cell killing to drive a sustained antitumour response.[23–25] Based on the different intrinsic mechanisms of action of NKR-2 observed in preclinical experiments on various tumour models, NKR-2 should overcome, at least partially, the local immunosuppression and target expression heterogenicity in tumour tissues, while, because of its effect on tumour vasculature, trafficking into tumours might also be enhanced. Furthermore, the multiple infusion strategy was also found to increase activity and

infiltration into tumours of other short-term persistent CAR T cells (mesothelin-specific RNA CAR T cells),[58] overcome the potential tumour-induced T cell dysfunction on interaction of CAR T cells with the immunosuppressive tumour microenvironment[23] while being well tolerated in humans.[59] Consequently, compared with classical CARs, clinical activity in solid tumours might be observed after treatment with NKR-2, without any construct modification nor combination with other immunotherapies. Since THINK is the first trial to evaluate the potential clinical activity of a multiple NKR-2 administration schedule in solid tumours, it offers a real opportunity compared with classical CAR T cells within the solid tumour-type field, where therapeutic options are limited. Further impacting the tumour environment through combinations of chemotherapy or immune modulation will constitute subsequent steps to consider once we have progressed sufficiently in the THINK trial and demonstrated safety of NKR-2 administrations.

The multiple infusion strategy, absence of preconditioning chemotherapy, diversity of putative mechanisms of action, as well as the diversity of tumour types that can potentially be targeted, therefore stand the NKR-2 CAR T cell approach described in this protocol apart from the current 'gold standard' CAR T cell therapy paradigm. In this context, the THINK trial provides a first approach of a global comprehensive clinical programme embedding other phase I clinical protocols that will evaluate the combination of the NKR-2 with specific anticancer treatments and the locoregional administration, among others.

### Ethics and dissemination
The investigator is responsible for ensuring the study is conducted in accordance with procedures and evaluations described in the clinical protocol. Deviations from the protocol shall not be made without prior approval by the sponsor, except to treat a medical emergency and reduce immediate risk to the patient. In such a case, the investigator shall immediately notify the independent ethics committee (IEC) or institutional review board and the sponsor, according to local requirements.

If changes to the protocol are deemed necessary, a written amendment should be prepared and signed by the investigator and the sponsor and approved by the IEC and applicable competent authorities prior to being implemented.

The *steering committee* will be composed of three members independent of investigators and the sponsor. The role of the steering committee will be to advise on events which may impact the general process of the clinical study. All safety data will be made available to the steering committee in a way that the steering committee can timely advise.

Celyad or designee will host specific *safety data review meetings* (or teleconferences) on a regular basis during the study to discuss and evaluate all of the gathered safety data. This safety follow-up will include teleconference at the end of each dose level of the phase I dose escalation arms and at any new relevant safety event onset (ie, DLT, treatment-related SAE, SUSAR). In addition, members of the steering committee will be requested to review the clinical study report(s) generated from data collected in this study.

*Clinical data review meetings* will be performed at the different study interim analyses for futility by the Celyad clinical trial team and the steering committee in specific. At the time of the interim analyses, the best overall response for each patient will be derived from the overall lesion response assessments recorded in the eCRF. This will be used to calculate the ORR in the respective group(s).

For the purpose of compliance with Good Clinical Practises (GCP) and competent authorities guidelines, it may be necessary for the sponsor or a competent authority to conduct a site audit. This may occur at any time from start to after conclusion of the study. Therefore, when an investigator signs the protocol, he agrees to permit drug competent authorities and the sponsor audits, providing direct access to source data/documents.

Written *informed consent* will be taken from all participants. The patient will be asked to read and review the document. On reviewing the document, the investigator (or delegate) will explain the research study to the patient and answer any questions that may arise. The patients consent should be appropriately recorded by means of the patients personally dated signature prior to any procedures being done specifically for the study. The patients should have the opportunity to discuss the study with their surrogates or think about it prior to agreeing to participate. The patient's participation is voluntary and the patient may withdraw consent at any time throughout the course of the study.

### Patient confidentiality
Patient confidentiality is strictly held in trust by the participating investigators, their staff, and the sponsor(s) and their agents. This confidentiality is extended to cover testing of biological samples and genetic tests in addition to the clinical information relating to participating patients.

The study protocol, documentation, data and all other information generated will be held in strict confidence. No information concerning the study or the data will be released to any unauthorised third party, without prior written approval of the sponsor.

The investigator remains accountable for the *study data*. Final data set will be stored in a central database maintained by the sponsor. At the conclusion of the study, the sponsor will archive the study data in accordance with internal procedures. All information provided by the sponsor and all data and information generated by the site as part of the study (other than a patient's medical records) remain the sole property of the sponsor.

The results of this study will be disseminated through presentation at international scientific conferences and reported in peer-reviewed scientific journals.

The sponsor holds and will maintain an adequate *insurance policy* covering damages arising out of Celyad's sponsored clinical studies. Financial compensation to investigators and/or institutions will be in accordance with the agreement established between the investigator and/or institutions and the sponsor.

The THINK trial (A Dose Escalation Phase I Study to Assess the Safety and Clinical Activity of Multiple Cancer Indications) is currently recruiting participants in Europe and USA.

**Author affiliations**
[1] Celyad SA, Mont-Saint-Guibert, Belgium
[2] Celyad SA, Boston, MA, USA
[3] Institut Jules Bordet, Université Libre de Bruxelles, Brussels, Belgium
[4] Cliniques Universitaires Saint-Luc, Université Catholique de Louvain, Brussels, Belgium
[5] International Drug Development Institute, Louvain-la-Neuve, Belgium
[6] H. Lee Moffitt Cancer Center, Tampa, Florida, USA
[7] Ghent University Hospital, Ghent, Belgium
[8] Roswell Park Cancer Institute, Buffalo, New York, USA
[9] Morsani College of Medicine, University of South Florida, Tampa, Florida, USA

**Acknowledgements** The delivery of successful clinical trials is primarily due to the excellence of the clinical partners with whom Celyad collaborates. The authors wish to thank all investigators in US and EU and all clinical managers for their valuable support and guidance. Finally, we thank all the patients and their families for participating to the study and for encouraging us to do this research.

**Contributors** CL, DEG and FFL conceived the study design, developed essential study documents and drafted the manuscript. BV, DEG and FFL participated in the coordination of the study. FP participated in the study design and the statistical review. AH, PA, AA, EVDN, GC, J-PHM, JBB, DAS, TK and KO participate in the management, will treat patients and conduct the clinical trial. MLD gave support and guidance during the study design conception. All authors read, reviewed and approved the final manuscript.

**Competing interests** CL, BV, DEG and FFL are employed by Celyad SA. JBB and PA declare no disclosures. J-PHM has advisory roles with Merck Sharp and Dohme (uncompensated), Innate Pharma, Debio, Boehringer-Ingelheim and Nanobiotix. The THINK clinical trial is sponsored by Celyad SA.

**Ethics approval** Ethical approval has been obtained at all sites open for recruitment. The trial has been approved by ClinicalTrials.gov (Trial registration ID NCT03018405) and by EU Clinical Trials Register (EudraCT number: 2016-003312-12). The present publication refers to protocol version 2.0 (released the 10th of February 2017).

**Provenance and peer review** Not commissioned; externally peer reviewed.

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
