## [Reviewer comments · BMJ Open]

ARTICLE DETAILS

TITLE (PROVISIONAL)	Study protocol for THINK: A multinational open-label Phase I study to assess the safety and clinical activity of multiple administrations of NKR-2 in patients with different metastatic tumour types
AUTHORS	Lonez, Caroline; Verma, Bikash; Hendlisz, Alain; Aftimos, Philippe; Awada, Ahmad; Van Den Neste, Eric; Catala, Gaetan; Machiels, Jean-Pascal; Piette, Fanny; Brayer, Jason; Sallman, David; Kerre, Tessa; Odunsi, Kunle; Davila, Marco; Gilham, David; Lehmann, Frédéric

VERSION 1 – REVIEW

REVIEWER	Rajesh Nandy University of North Texas Health Science Center USA
REVIEW RETURNED	22-May-2017
GENERAL COMMENTS	Since this a proposed Phase I study, no actual data analysis can be performed. However, the proposed design is adequate.

REVIEWER	Robert Canter UC Davis, USA
REVIEW RETURNED	27-Jul-2017

GENERAL COMMENTS	Interesting concept for a clinical trial. Background section on significance of and rationale for NKG2D CAR therapy is good. Methods section describes structure of the trial well. Discussion is a little superficial, especially with respect to expected outcomes for trial and correlative studies as well as how the investigators will address barriers to successful CAR therapy in solid tumors which the authors acknowledge have limited efficacy of CAR therapy in previous trials, namely trafficking to solid tumors, local immune suppression, tumor heterogeneity, and off-tumor effects (on-target). These are anticipated to limit efficacy of authors' trial, and there is no discussion of how these barriers will be overcome. Additional comments: 1) Why is 2-year period put forward for assessment of best response to treatment? If patients progress with experimental treatment and then receive other cancer therapy for salvage, how will these salvage therapies potentially impact response assessment for NKR-2 treatment?
---

	2) How do authors propose to manage potential toxicity from binding to NKG2D ligands in non-neoplastic host tissue? This could be a significant source of toxicity, and excluding patients with an infectious/inflammatory insult in the recent weeks before trial registration may screen out some patients at risk for on-target, off-tumor side effects, how will patients be managed who develop toxicity while on treatment? It is not clear at all that anti-IL6 treatment (as for CRS) will work in this case.
--	--

VERSION 1 – AUTHOR RESPONSE

Reviewer: 1

Comment: Since this a proposed Phase I study, no actual data analysis can be performed. However, the proposed design is adequate.

Response: We would like to thank the reviewer for taking time to review our work.

Reviewer: 2

Comment: Interesting concept for a clinical trial. Background section on significance of and rationale for NKG2D CAR therapy is good. Methods section describes structure of the trial well.

Response: We thank the reviewer for his comments and careful reviewing of our work.

Comment: Discussion is a little superficial, especially with respect to expected outcomes for trial and correlative studies as well as how the investigators will address barriers to successful CAR therapy in solid tumors which the authors acknowledge have limited efficacy of CAR therapy in previous trials, namely trafficking to solid tumors, local immune suppression, tumor heterogeneity, and off-tumor effects (on-target). These are anticipated to limit efficacy of authors' trial, and there is no discussion of how these barriers will be overcome.

Response: The reviewer is right. Until now “classical” CAR T-cell therapy has been confronted to several hurdles in the field of solid tumours, which may explain the overall lack of positive clinical results from previous clinical trials.. The aim of the discussion, as well as background section, was to explain that, based on the several intrinsic mechanisms of action and multiple targeting ability of NKR-2, our CAR-T might reach clinical activity in solid tumours as stand-alone, i.e., without any combination with other therapeutics or adding other specific features into the chimeric construct to improve the intra-tumour trafficking and/or to target the local immune suppression. Indeed, based on preclinical data, NKR-2 should overcome, at least partially, local immunosuppression and target expression heterogeneity in tumour tissues, and because of its effect on tumour vasculature, trafficking into tumours will be also enhanced. This trial is the first trial to evaluate the potential clinical activity of a multiple NKR-2 administration schedule in solid tumours, so the aim was to assess if the single “agent” NKR-2 might be effective as such and, based on results and exploratory research, potentially design a later trial to favour NKR-2 activity. Interesting to note, THINK study is also evaluating in parallel 2 hematologic indications (AML and MM), in which the clinical and safety endpoints will be evaluated separately from the solid indications based on these aspects, i.e., the direct tumour recognition in the absence of important trafficking to tumour need may lead to another “target product profile” in terms of clinical activity and adverse events profile.

Please note that THINK study is the first study of a global comprehensive clinical program embedding other phase I clinical protocols that will evaluate the combination of the NKR-2 with specific anti-cancer treatments and the loco-regional administration, among others.

For your information, one future trial (recently accepted by authorities and EC) will assess the NKR-2 activity in patients with unresectable liver metastases, when NKR-2 is administered locally via hepatic arterial infusion to increase further the trafficking of NKR-2 into tumours and decrease potentially systemic toxicity. Another trial (also recently accepted) will evaluate NKR-2 safety and clinical activity when administered concurrently with a standard chemotherapy regimen to patients with potentially resectable liver metastases. These trials results will then be analysed to define the best therapeutic option to assess in future clinical development.

All these aspects have been clarified in the completely revised version of the discussion, as proposed by the reviewer.

Additional comments:

1) Why is 2-year period put forward for assessment of best response to treatment? If patients progress with experimental treatment and then receive other cancer therapy for salvage, how will these salvage therapies potentially impact response assessment for NKR-2 treatment?

Response: The main treatment follow-up will indeed be performed until 2 years after 1st NKR-2 administration. Later on, a long-term safety follow-up will follow the patients up to 15 years after protocol initiation.

Obviously, the assessment of the response during the first 2 years will be made at regular times, as depicted in figure 4. First tumour assessment will be performed 2 months after 1st NKR-2 administration (D57), then at M3, M6, M9, M18 and M24 and at any time if considered as needed per investigator's decision (e.g. progression suspicion) and in regard to the RECIST 1.1. response confirmation. Endpoints include all these timepoints, and data analysis will not be limited to the M24 timepoint to assess best response to treatment.

For all the patient population targeted by this protocol, corresponding to standard phase I patient population with refractory or relapsing patients post standard metastatic treatment line(s), an absence of progression after the first two years post treatment initiation would be considered as a significant sign of clinical activity.

In case of absence of disease progression at M24, PFS and OS as clinical endpoints will be followed-up on a yearly basis or at ad hoc basis, i.e. in case of progression/relapse suspicion. In case of disease progression at M24 or at any time before, the patient will be still asked to attend simplified follow-up procedures which will include (i) record of survival status, (ii) medication status and (iii) blood sampling for Viral copy number (VCN) and RCR assay, as requested by FDA.

Salvage therapies are only accepted per protocol in case of clinical progression. The PFS information will be therefore obtained at the time of documented progression. The question of the potential impact of any salvage therapy on the response assessment and most specifically on the overall survival induced by the initial investigational NKR-2 treatment will depend of some putative mechanisms of action still to be evaluated in human into the present study protocol but also of the type of salvage therapy. As an example, if NKR-2 therapy induces a specific anti-tumour immune response by itself, will salvage chemotherapy impair this immune response or will a salvage T-cell checkpoint therapy positively impact this immune response?

This was clarified in the amended version.

2) How do authors propose to manage potential toxicity from binding to NKG2D ligands in non-neoplastic host tissue? This could be a significant source of toxicity, and excluding patients with an infectious/inflammatory insult in the recent weeks before trial registration may screen out some patients at risk for on-target, off-tumor side effects, how will patients be managed who develop toxicity while on treatment? It is not clear at all that anti-IL6 treatment (as for CRS) will work in this case.

Response: The toxicity related to expression of NKG2D ligands in normal host tissue, especially in the context of infectious/inflammatory insult, may indeed be a cause of toxicity exacerbation, and potentially induce a cytokine release syndrome due to the increased recognition of ligands by NKR-2 cells. This syndrome will be managed following the standard procedures as written in the toxicity management section.

CRS will be potentially treated by anti-IL-6 treatment (e.g. Tocilizumab) or other immunosuppressive therapy (e.g. corticosteroids) per “standard” rules as set for classical CAR-T approach. Please note that it is requested and monitored locally that all participating institutions do have tocilizumab available in house to manage such CRS toxicity. These rules have been reviewed by the study IDMC and are identical that the one used by other companies developing CAR-Ts.

Other toxicity may however arise from the damaging of the host tissues after their recognition by NKR-2.

As uncontrolled expansion and proliferation of the transformed cells facilitates the occurrence of these toxicities, the persistence of NKR-2 will be carefully monitored. Indeed, even if, in contrast to more classical CAR T therapies, NKR-2 do not have significant in vivo expansion (both in preclinical and clinical studies) nor long term persistence, this is the first study with NKR-2 administered as multiple doses, and this might potentially modify its toxicity profile. Therefore, in case off uncontrolled T-cell proliferation occurrence (grade 3 or 4 toxicity related to NRK-2), and since CAR associated toxicity has been reported to respond to systemic corticosteroids, patients may be treated with corticosteroids and/or chemotherapy to eradicate the CAR T-cells.

Finally, as the reviewer comments, infection and/or inflammation could indeed increase the NKG2D ligand expression, hence potential toxicity. Patients with active infection or inflammatory auto-immune disease will be therefore excluded from the study at screening. Eligibility criteria are strict on that sense as well all sites participating have been specifically informed about this protocol aspect.

Furthermore, and as written in the toxicity management section, in case of acute infection, as well as fever of 38°C/100°F or above, will be also considered a postponement criterion when observed during the treatment phase. In this case, all study procedures and treatment administration will be resumed where it was left as soon as the patient’s condition allows (within 4 weeks after the scheduled treatment date) or at the investigator’s discretion after Steering Committee and Sponsor approvals. Toxicity management and eligibility criteria sections have been modified to highlight more clearly this aspect.

VERSION 2 – REVIEW

REVIEWER	Robert Canter UC Davis, USA No Competing Interest
REVIEW RETURNED	29-Aug-2017
GENERAL COMMENTS	No further comments